# Review: Prevalence of Addictions among Transgender and Gender Diverse Subgroups

**DOI:** 10.3390/ijerph18168843

**Published:** 2021-08-22

**Authors:** Ryan Ruppert, Shanna K. Kattari, Steve Sussman

**Affiliations:** 1Keck School of Medicine of USC, Los Angeles, CA 90033, USA; ssussma@usc.edu; 2School of Social Work, University of Michigan, Ann Arbor, MI 48109, USA; skattari@umich.edu

**Keywords:** addiction, substance use disorder, behavioral addiction, LGBTQ, transgender, gender diverse

## Abstract

We conducted an analysis of the prevalence of substance and behavioral addictions across different transgender and gender diverse (TGD) subgroups. We performed a scoping review using MEDLINE and Google Scholar databases and examined 12 addictions, including alcohol, nicotine, cannabis, illicit drugs, gambling, eating/food, internet, sex, love, exercise, work, and shopping. We presented prevalence rates for each addiction as a function of an individual’s gender identity (stratified into transgender females, transgender males, and gender nonconforming), and used cisgender women and men as reference groups. We included 55 studies in our final analysis, the majority of which investigated substance use disorders among TGD subgroups. Overall findings indicated that substantial differences in substance use exist among US TGD subgroups. There were far fewer publications that examined the prevalence of behavioral addictions across TGD subgroups. However, despite limited research in this area, findings still suggest that notable differences in behavioral addictions may exist between individual TGD subgroups. The conclusions of our review may provide clinicians with a better ability to screen for and treat at-risk individuals within the TGD community.

## 1. Introduction

Despite growing acceptance of the lesbian, gay, bisexual, transgender, and queer (LGBTQ) community over recent years, this subset of the population remains at a significantly elevated risk of addiction compared to the general U.S. adult population [1]. While many studies have historically analyzed sexual minorities (SMs) and gender minorities (GMs) as single homogenous groups, there is a growing recognition of the significant differences that exist among individual LGBQ and T subgroups. In particular, it has been shown that transgender and gender diverse (TGD) populations face substantially greater healthcare barriers and poorer health outcomes than their cisgender SM counterparts. For instance, a nationwide study determined that 40.4% of TGD adults had attempted suicide during their lifetime, compared to 17% of cisgender SMs, and only 2.4% of the general U.S. population [2]. TGD individuals are also four times more likely to experience poverty, four times more likely to have HIV, and twice as likely to be unemployed in comparison to cisgender adults [3]. Additionally, it has been shown that TGD individuals and communities experience high rates of discrimination and low levels of social support, two factors which are believed to play a mediating role in the development of addictive behaviors [4,5,6]. Research has also indicated that substance use within the TGD community is more associated with sexual risk and mental health complications, further suggesting that this subset of the LGBTQ community is particularly vulnerable to the deleterious effects of addiction [7].

A recent review analyzed the prevalence and co-occurrence of a wide range of substance use disorders and behavioral addictions among four cisgender SM subgroups (i.e., gay men, lesbian women, bisexual men, and bisexual women) and concluded that addiction rates varied markedly from one subgroup to another [1]. In this paper, we have applied a similar framework, but with the aim of better understanding how addiction rates may differ between individual TGD subgroups. While we have included a number of studies which outline addiction disparities between TGD and cisgender populations, we primarily focused on research that compared addiction rates between transgender females (an individual who was designated male at birth but who identifies as female) and transgender males (an individual who was designated female at birth but who identifies as male). We also identified a number of recently published studies that stratified populations into gender nonconforming (GNC) or gender nonbinary (GNB) subgroups (umbrella terms for gender identities that are neither exclusively men nor women). Research indicates that GNC and GNB subgroups, despite being relatively understudied, may represent a particularly vulnerable group even within the high-risk TGD community [8].

In this review, as in previous studies, we analyzed the prevalence of 12 addictions (substances: alcohol, nicotine, marijuana, and illicit drugs; behaviors: gambling, food/eating, Internet, exercise, sex, love, shopping, and work) [1,9]. We have prioritized studies that assessed addictions using DSM-V criteria; however, we included several studies which measured addictions using alternative metrics (e.g., hazardous drinking as a proxy for an alcohol use disorder). When available, we presented results for each addiction based on total U.S. prevalence as well as prevalence within up to four stratified GM subgroups (i.e., transgender females, transgender males, nonbinary assigned female at birth [AFAB], and nonbinary assigned male at birth [AMAB]). When quantitative data were unavailable, we incorporated a selection of non-prevalence studies examining factors that may predispose certain TGD subgroups to develop an addictive behavior. Finally, we pinpointed specific areas of this field that warrant further research and presented considerations aimed at assisting clinicians in their efforts to identify high-risk individuals within the TGD community.

## 2. Methods

The search involved using MEDLINE and Google Scholar databases to investigate studies published before March 2021. Search terms included: (Transgender OR “GM” OR LGBT OR LGBTQ OR “Gender Nonconforming” OR “Non-binary”) AND (prevalence OR incidence) AND (co-occurrence OR comorbidity OR “co-occurring disorders” OR “co-occurring addictions”). We included terms for each addiction subtype referenced in Sussman et al. plus pertinent terms from the Diagnostic Statistical Manual of Mental Disorders—Fifth Edition (DSM-V): (i) Alcohol: “alcohol dependence”, “alcohol use disorder”, alcoholism; (ii) Tobacco: “tobacco addiction”, “tobacco use disorder”, “nicotine addiction”, “nicotine dependence”, (iii) Marijuana: “marijuana abuse”, “marijuana dependence”, “marijuana use disorder”, “cannabis dependence”, “cannabis use disorder”, (iv) Illicit substances: “illicit drug abuse”, “drug abuse”, “drug dependence”, “drug addiction”, “substance use disorder”, (v) Gambling: “gambling addiction”, “compulsive gambling”, “pathological gambling”, “gambling disorder”, (vi) Eating: “overeating addiction”, “food addiction”, “eating addiction”, “binge eating disorder”, “overeating dependence”, “eating disorder”, “feeding disorder”, (vii) Internet: “internet addiction”, “web addiction”, “pathological internet use”, “video game addiction”, (viii) Love: “love addiction”, “pathological love”, (ix) Sex: “sex addiction”, “sexual compulsivity”, “sexual dependency”, (x) Exercise: “physical activity addiction”, “exercise addiction”, (xi) Work: “workaholic”, “workaholism”, “work addiction”, and (xii) Shopping: “shopping addiction”, “compulsive shopping”. [9,10,11] A total of six TGD-related terms were crossed with a total of eight prevalence/incidence-related terms and crossed with 44 addiction-related terms, across two search engines, leading to a total of 4224 searches (6 ×8 × 44 × 2).

As detailed in Figure 1, we located 643 articles after applying the aforementioned search terms, including studies that included these terms in the title, abstract, or body of the manuscript. We also examined reference sections to locate additional publications not identified through database searches; this resulted in the inclusion of 19 additional studies. Of the 662 identified articles, 117 duplicate articles were removed, leaving a total of 545 papers.

We established the following inclusion and exclusion criteria in order to yield a wide range of publications with a degree of comparability across manuscripts:The study samples included adult patients (defined as 18+ years of age) who identified as either transgender, gender nonbinary, and/or gender nonconforming.Studies addressed the prevalence of at least one SUD or behavioral addiction; proxies for addictions (i.e., heavy episodic drinking for AUD) were also considered.Studies included more than one TGD sub-population in the analysis and/or a cisgender reference group.Studies were peer-reviewed.Studies were written in English.

We also gave priority to studied with sample sizes that exceeded 500 participants; however, if few studies were located for a particular addiction, we allowed for the inclusion of studies with smaller sample sizes. We also aimed to exclusively include studies conducted in the United States, but given the limited number of relevant studies, we considered non-U.S. publications as well. A total of 89 studies were retained after applying the aforementioned criteria. An additional four studies were added after analyzing the reference lists of the 89 retained papers. Among the remaining 93 eligible studies, 38 more studies were removed for reasons listed in Figure 1, which resulted in 55 manuscripts that were included in our review.

## 3. Results

### 3.1. Alcohol Use Disorders

There are a number of studies which have investigated the prevalence of alcohol use disorders (AUDs) among TGD adults. Many of these studies, which analyze TGD participants as a single, non-stratified group, suggest that this population is at a markedly elevated risk of binge drinking, which is considered a proxy for the future development of an AUD [12]. In fact, a study of 452 transgender adults concluded that 47% of participants reported an episode of binge drinking within the previous 3 months [13]. For comparison, the prevalence of binge drinking among the general U.S. population has been estimated at 17.1% [14]. Another study, which analyzed the prevalence of past month binge drinking among 406 GMs, concluded that transgender participants were nearly 50% more likely to report past month binge drinking compared to their cisgender counterparts [15]. This disparity is further evidenced by an investigation of drinking patterns among 335 transgender young adults [16]. This particular study determined that 26.96% of transgender participants reported past-month heavy episodic drinking (HED) compared to only 8.57% of cisgender participants, suggesting that transgender individuals in this age group may be over three times more likely than their cisgender counterparts to engage in higher-risk alcohol consumption.

In contrast to the aforementioned findings, we also located several studies indicating that the risk of developing an AUD is similar between cisgender and transgender populations. An analysis of the 2014 Behavioral Risk Factor Surveillance System concluded that 16.2% of cisgender participants reported a past-month HED compared to 11.5% of transgender participants [17]. While mean percentage prevalence was higher among cisgender participants, there was no statistically significant difference between the two groups. Another study, which analyzed a group of 350 transgender adults, concluded that 23% of participants reported a lifetime AUD [18]. While this study did not include a cisgender reference group, it should be noted that the National Epidemiologic Survey on Alcohol and Related Conditions III determined that 29.1% of the general U.S. population reported a lifetime AUD [19]. In another study, which analyzed 175 transgender adults between the ages of 18–29, it was concluded that cisgender men had the highest rates of HED at 40.9% followed by cisgender women at 28.1%, and transgender participants at 27.4% [20]. However, in addition to measuring HED prevalence, the researchers in this study also examined the rate of alcohol-related problems as a function of gender identity and concluded that transgender individuals were at a significantly elevated risk. For instance, 7.2% of transgender participants reported a prior episode of alcohol-related sexual assault compared to only 2.1% of cisgender women and 1.1% of cisgender men, though it should be noted that TGD individuals are already at significantly higher risk for sexual assault than cisgender individuals regardless of alcohol intake [21]. Additionally, this study found that 12.2% of transgender individuals reported suicidality related to alcohol consumption compared to only 2.1% of cisgender men and 1.9% of cisgender women. Despite varying outcomes regarding the prevalence of AUDs between cisgender and transgender populations, there is convincing evidence to suggest that transgender individuals are more likely to experience alcohol-related problems compared to their cisgender counterparts.

#### 3.1.1. Transgender Men

While the aforementioned studies analyze transgender individuals as a single group, there is a growing body of research aimed at investigating how drinking patterns may vary between individual TGD subgroups. Several studies suggest that transgender men (TM) may be at an elevated risk of developing an AUD compared to transgender women (TW). A study of 433 transgender adults found that 42.2% of TM reported HED within the previous year compared to 22.7% of TW participants [22]. When compared to age-adjusted cisgender reference groups, TM reported higher rates of HED than both cisgender men and women. While TW were reported to have higher rates of HED than age-adjusted cisgender women, their rates were noted to be lower than age-adjusted cisgender men. Another study measured alcohol consumption by assessing current alcohol use, current binge drinking, and current frequent binge drinking among a group of 27,715 U.S. GM adults [23]. This study found that TM reported higher rates across all three measures with 63.0% of TM reporting current alcohol use (vs. 57.6% of TW), 26.8% of TM reporting current binge drinking (vs. 21.5% of TW), and 8.1% of TM reporting current frequent binge drinking (vs. 7.2% of TW). In a 2019 study, Tomita et al. investigated whether transgender people who had received gender affirming medical interventions (GAMIs) would report different behavioral health outcomes compared to transgender people who desired but had not engaged in GAMIs [24]. While there were no significant changes in health outcomes among TW in the sample, the investigation did conclude that TM who received GAMIs reported lower scores on depression, social anxiety, generalized anxiety disorder, and PTSD compared to those who had not received any type of intervention. Interestingly, though, the TM who received GAMIs also scored significantly higher in terms of alcohol abuse compared to the control group. Despite showing improvements in a wide range of mental health outcomes, the TM in this sample were found to be uniquely vulnerable to alcohol abuse.

#### 3.1.2. Transgender Women

We also located a number of studies investigating alcohol use patterns among TW. While much of this research characterized the prevalence of TW alcohol use without the inclusion of a reference group, there are several recent studies investigating how TW may face a higher risk of AUDs in comparison to other groups [7,25,26,27,28,29]. For instance, a study of 989 transgender young adults found that 66.7% of TW reported alcohol consumption within the previous two weeks; this rate exceeded that of cisgender men (62.3%), TM (57.9%), and cisgender women (55.0%) [30]. In addition to assessing for alcohol use prevalence, this study also examined other metrics of alcohol abuse as a function of gender identity. For instance, it was determined that TW reported significantly higher rates of alcohol-related blackouts compared to TM and both cisgender reference groups. This study also measured 21 negative alcohol-related consequences (i.e., engaging in physical altercations, experiencing sexual assault, etc.). When TW and TM were compared, it was determined that TW had statistically higher rates for 19 out of the 21 total alcohol-related consequences, indicating that TW with AUDs may be particularly vulnerable to the negative effects of drinking.

#### 3.1.3. Nonbinary Individuals

We also identified a number of recently published studies that examined alcohol use patterns among three subgroups of GMs: TW, TM, and GNC individuals. A 2019 study of 3063 GMs determined that individuals who identified as GNC reported the highest levels of current binge drinking at 22.7%, followed by TW (18.8%), cisgender individuals (16.6%), and TM (12.9%) [31]. Another study determined that GNC individuals reported a significantly higher rate of drinks per week (13.33) compared to both TW (9.18) and TM (6.22) [32]. This study also measured average AUDIT (a screening tool used to assess for hazardous drinking) scores among the three subgroups and determined that GNC individuals reported higher scores than either of the two transgender subgroups. 

We also located two recent studies which further stratified TGD individuals into separate groups based on their assigned sex at birth: Nonbinary Assigned Male at Birth (NBAMAB) individuals and Nonbinary Assigned Female at Birth (NBAFAB) individuals. One study measured the prevalence of frequent binge drinking among GM subgroups and determined that NBAMAB individuals ranked the highest at 14.9% followed by TM (8.1%), TW (7.2%), and NBAFAB individuals (5.8%) [23]. Of note, the prevalence rate reported by the NBAMAB subgroup was over 80% higher than the TM subgroup and over twice as high as both the TW and NBAFAB subgroups. The same study also assessed for the prevalence of current alcohol use and again determined that NBAMAB individuals reported higher rates than the other three subgroups. Another study, which measured the average AUDIT scores among TGD subgroups, found that NBAMAB individuals reported the highest average scores at 7.89, followed by TW at 5.52, TM at 3.70, and NBAFAB individuals at 3.44 (F = 5.26, *p* = 0.002) [33]. Despite utilizing different metrics to assess for AUD risk, the two aforementioned studies concluded that NBAMAB individuals may be at a higher risk compared to other GM subgroups.

### 3.2. Nicotine Use Disorders

There are numerous studies indicating that the prevalence of smoking is markedly elevated among members of the transgender community [18,34]. A 2019 study by Wheldon et al. found that 34.9% of transgender participants reported current use of cigarettes compared to 22.4% of cisgender individuals (*p* = 0.003) [35]. Another study determined that transgender young adults, when compared to their cisgender counterparts, were 2.7 x more likely to report lifetime cigarette use and 4.2 x more likely to report cigarette use within the previous 30 days [16]. In addition to examining cigarette use, recent studies have also focused on other forms of nicotine consumption (i.e., e-cigarettes and smokeless tobacco) as a function of gender identity. A 2017 study measured the prevalence of both cigarette and e-cigarette use among a sample of transgender and cisgender adults [25]. Compared to their cisgender counterparts, transgender individuals were over 70% more likely to report cigarette use within the previous 30 days (*p* = 0.001). However, when e-cigarette consumption was measured, transgender individuals were over four times more likely to report use compared to cisgender participants (*p* < 0.0001). This disproportionately elevated risk of e-cigarette use among transgender populations has been demonstrated in several other studies, including a 2019 study in which the prevalence of current e-cigarette use was over 90% higher among transgender participants compared to the cisgender reference group [17,36]. Another study, which analyzed 3063 transgender adults, determined that the prevalence of current smokeless tobacco use among transgender individuals was over two times higher than the rate reported by cisgender individuals [31].

#### 3.2.1. Transgender vs. Cisgender Sexual Minorities Subgroups

Studies have also investigated how smoking rates among transgender populations compare to their cisgender SM counterparts. A 2019 study by Delahanty et al. examined the prevalence of nicotine use among a sample of 3898 SM stratified into four cisgender SM subgroups (i.e., cisgender gay men, cisgender bisexual men, cisgender lesbian women, and cisgender bisexual women) and a single GM subgroup consisting of TW, TM, and GNC individuals [37]. This study determined that the GM subgroup reported the highest prevalence of past month cigarette use compared to the other four cisgender SM subgroups. However, when the prevalence of past month e-cigarette use was measured, GMs ranked fourth behind cisgender bisexual men, cisgender bisexual women, and cisgender lesbian women. Another study, which analyzed a sample of 4159 adults between the ages of 18–34, stratified participants into three SM/GM subgroups (lesbian/gay, bisexual, and transgender) and a single cisgender heterosexual reference group [38]. While the heterosexual group reported the lowest rate of current smoking at 22.2%, the three SM/GM subgroups reported similarly elevated smoking rates; lesbian/gay participants ranked highest at 34.8%, followed by transgender individuals at 33.2%, and bisexual individuals at 31.1%. While the majority of these studies suggest that transgender individuals are at a higher risk of NUDs compared to the general cisgender population, the current literature has yet to establish whether the prevalence of smoking among transgender individuals is significantly different from their cisgender SM counterparts.

#### 3.2.2. Transgender Men

We identified several studies which compared NUD risk between individual TGD subgroups. Most of these studies suggested that TM may be at a higher risk than TW. For instance, results from the 2015 U.S. Transgender Survey indicated that TM, when compared to TW, were 34% more likely to report current use of cigarettes (*p* < 0.001), 24% more likely to report current e-cigarette use (*p* < 0.001), and 18% more likely to report dual use of both cigarettes and e-cigarettes (*p* < 0.05) [2]. We also identified several studies that strongly suggested that TM were at an elevated risk of NUDs relative to TW; however, due to small sample sizes, the results of these studies did not reach statistical significance. For example, a study of 677 transgender young adults found that 31% of TMs endorsed daily smoking compared to 23.3% of TFs [39]. Another study of TGD adults concluded that 17.2% of TM reported a current NUD compared to 13.5% of TW, and 11.2% of cisgender individuals [31].

#### 3.2.3. Transgender Women

While most of the studies exploring the prevalence of NUDs among TW populations did not include a TGD reference group, we did identify one study which indicated that TW were at a higher risk than TM [28,40]. This study, which analyzed TGD individuals living in Chicago, determined that 41.4% of TW participants reported current smoking compared to only 6.8% of TM [33]. Despite this significant difference in smoking prevalence between TW and TM, it should be noted that this particular study not only sampled a relatively small number of TGD participants (*n* = 214), but also specifically analyzed individuals between the ages of 16 and 32.

#### 3.2.4. Nonbinary Individuals

The current literature indicates that nonbinary populations are not at a particularly elevated risk of smoking when compared to their TW and TM counterparts. A recent study of TGD young adults concluded that 19.2% of nonbinary participants reported current use of cigarettes, which ranked lower than both TM (28.2%) and TW (23.8%) [37]. When assessing for e-cigarette use, this study determined that 9.1% of nonbinary individuals reported current use compared to 12.5% of TM and 8.8% of TW. A 2019 study of 3063 TGD individuals determined that nonbinary participants reported lower prevalence rates of both current cigarette and e-cigarette smoking when compared to TW and TM [31]. Another study, which stratified nonbinary individuals by their sex assigned at birth, concluded that NBAFAB individuals were significantly less likely to endorse current smoking compared to TW, while NBAMAB smoking rates were not statistically different from any of the TGD subgroups [33].

### 3.3. Cannabis Use Disorders

Several recent studies suggest that cannabis use is markedly elevated within TGD communities, with one study determining that over 40% of transgender participants reported current cannabis use [35,41]. This elevated risk of developing a cannabis use disorder (CUD) is further highlighted by a number of studies which directly compare cannabis use rates between cisgender and TGD groups. In fact, one study of young adults based in California found that the prevalence of lifetime cannabis use was nearly two times higher among transgender participants compared to cisgender individuals [16]. This study also calculated the prevalence of past month cannabis use and concluded that 29.47% of the transgender group reported past month cannabis use compared to 11.56% of the cisgender reference group (AOR = 1.93, 95% CI: 1.35–2.75, *p* < 0.001). A 2018 study of 406 transgender and GNC adults based in Colorado found that 32.8% of GM participants reported current cannabis use compared to only 13.6% of the statewide cisgender adult population [15].

#### 3.3.1. Transgender Men

We located only a limited number of studies which directly compared cannabis use between TGD subgroups; however, those studies that did stratify transgender participants suggested that TM may be at a higher risk for CUDs compared to TW. For instance, a study of 1210 transgender U.S. adults found that the prevalence of cannabis use within the previous three months was 31.3% among TM compared to 19.0% among TW (*p* < 0.001) [42]. Another study, which calculated rates of cannabis use among 201 transgender adults, determined that the prevalence of current CUDs was 39% higher among TM compared to TW [43].

#### 3.3.2. Transgender Women

We also located several studies which explored cannabis use among TW; however, the majority of these studies did not include cisgender or TGD reference groups [25,26,33]. One of these studies longitudinally followed the same group of TW over 17 years and determined that the prevalence of cannabis use significantly increased over time. In fact, 54.2% of TFs reported past six-month cannabis use between 2015–2016 compared to 38.9% of the same sample between 1998–1999 (*p* < 0.005) [44]. Of note, this study also longitudinally measured several other substance use rates (including alcohol, cocaine, and methamphetamine) among TW and determined that cannabis use was the only measurement that significantly increased over time, and it should also be noted the legalization of cannabis has occurred in several states over this same time period.

#### 3.3.3. Nonbinary Individuals

We only identified one study which assessed cannabis use among nonbinary subgroups. This particular study, which analyzed 214 TGD adults, concluded that 39.3% of NBAMAB participants reported current hazardous marijuana use, which exceeded the rates reported by TW, TM, and NBAFAB individuals [33]. This study also reported that NBAMAB individuals reported higher CUDIT scores (an assessment tool for measuring problematic cannabis use) than their other TGD counterparts, though statistical significance was not reached.

### 3.4. Illicit Drugs

There is compelling research to suggest that the prevalence of illicit drug use is markedly elevated among TGD communities [45,46]. A recent study determined that 24.68% of transgender young adults reported past month illicit drug use, which was nearly five times higher than the 5.11% of cisgender participants who reported similar use [16]. Another study, which stratified a sample of SM based on their gender identity, concluded that transgender SMs and cisgender SMs reported nearly identical rates of current drug use patterns [47]. For instance, it was determined that 70.3% of transgender SM participants reported current drug use compared to 71.6% of cisgender SMs; however, it should be noted that this study combined illicit drugs and cannabis into a single category, which likely explains the significantly elevated prevalence rates in both groups.

#### 3.4.1. Transgender Men

The literature regarding illicit drug use among TM is quite limited. We identified two studies indicating that TM may be at an elevated risk of illicit drug use compared to their TW counterparts. A study of 1210 transgender adults concluded that TM were at a significantly elevated risk of past 3-month illicit drug use compared to TW (12.9% vs. 10.6%; *p* < 0.001) [42]. Another study, which analyzed substance use patterns among a sample of 1229 TGD adults, found that the prevalence of illicit drug use was higher among TM (11.6%) than TW (8.9%); however, this study did not report whether the difference between subgroups reached statistical significance [48].

#### 3.4.2. Transgender Women

There is a strong body of research investigating the prevalence of illicit drug use among the TW community [49,50,51]. Although many of these studies examine TW drug use without the context of a reference group, we located a number of studies that stratified participants into two transgender subgroups. One study, which measured lifetime injection drug use among 292 transgender adults, concluded that 17.0% of TW participants reported past use compared to only 4.4% of TM participants (*p* < 0.05) [52]. Another study, which measured the prevalence of past year drug use as a function of gender identity, determined that TW were significantly more likely to report past year crack cocaine use compared to TM (4.5% vs. 0.7%) [53]. However, this study also measured the prevalence of various other drug types (i.e., ketamine, GHB, heroin, etc.) and concluded that no significant differences existed between the two transgender subgroups.

Throughout our literature search, we identified numerous studies suggesting that methamphetamine use may be particularly elevated in the TW community. For instance, a recent study determined that the prevalence of past 6-month stimulant use (i.e., methamphetamine and cocaine) was over three times higher among TW compared to TM (20.7% vs. 6.8%) [33]. Additionally, a longitudinal study of 271 TW concluded that participants were significantly less likely to use alcohol, cannabis, cocaine, and crack in 2015–2016 versus 1998–1999; however, the prevalence of methamphetamine use remained virtually unchanged over time (27.9% vs. 27.3%; *p* = 0.965) [44]. Research has also indicated that methamphetamine use in the TW community may be particularly associated with sexual risk factors. For instance, a study of 314 TW adults determined that 20.1% of participants reported past year methamphetamine use; of these participants, over 80% acknowledged that their methamphetamine use was performed either before or during anal intercourse [25]. Another study, which examined substance use patterns among a group of 2136 TW adults, determined that past month methamphetamine use was significantly higher among HIV+ women compared to HIV- women (29.2% vs. 20.3%; *p*< 0. 001) [26]. The same study also found that alcohol use was significantly higher in the HIV- group, further indicating that methamphetamine use may be uniquely associated with increased sexual risk.

#### 3.4.3. Nonbinary Individuals

We only identified one study that examined illicit drug use among nonbinary subgroups [33]. This particular study analyzed illicit drug use among four GM subgroups: TW, TM, NBAFAB individuals, and NBAMAB individuals. Overall, the results of this study suggest that nonbinary individuals are not at a particularly heightened risk of illicit drug use compared to their TW and TM counterparts. In fact, it was concluded that NBAFAB participants were significantly less likely to report stimulant use (i.e., cocaine and/or methamphetamine) compared to TW in the sample (2.4% vs. 20.7%). This study also measured the rate of club drug use (i.e., GHB, ketamine, and/or ecstasy) and ‘other drug’ use (i.e., heroin, inhalants, hallucinogens, and/or psychedelics) and found no statistical differences in prevalence between any of the four TGD subgroups.

### 3.5. Behavioral Addictions

There is a limited but revealing body of research investigating behavioral addictions as a function of gender identity. For instance, a 2019 study of 2168 transgender young adults determined that the prevalence of past year gambling was similar between transgender and cisgender populations (29.6% vs. 31.7%) [54]. However, when participants were assessed for pathological gambling (PG), it was determined that, while still very low, the prevalence of PG was nearly three times higher among TGD people compared to cisgender individuals (1.43% vs. 0.49%). When participants were further stratified, it was revealed that TW and cisgender men reported similar rates of past year gambling (43.5% vs. 43.1%), as did TM and cisgender women (24.0% vs. 20.2%). However, when PG was measured, it was found that TW had the highest prevalence at 3.07%, followed by cisgender males (0.79%), TM (0.69%), and cisgender women (0.17%). Despite transgender and cisgender individuals gambling at similar rates, this study strongly suggests that TGD individuals, specifically TW, may be at a heightened risk of PG.

Food addiction has also been examined among TGD communities, with one study estimating that 20.5% of transgender individuals have experienced a food addiction within their lifetime [55,56,57]. In a 2017 study, Lipson et al. measured the prevalence of past month binge eating (a proxy of food addiction) among a sample of college students and found that cisgender women reported the highest rate at 49.09%, followed by transgender participants (36.33%), and cisgender men (29.99%) [58]. It is important to note that while there are theoretical differences between binge eating and food addiction, studies indicate that the central features of these two behaviors are strongly linked (i.e., compulsive eating, excessive consumption despite adverse consequences, and diminished self-control overeating behaviors) [59]. In the same study, Lipson et al. also concluded that there were no significant differences in binge eating rates between TW and TMs, which has been supported by other studies [60]. We also identified one study which analyzed past year binge eating among TW, TM, and GNC subgroups. Though subgroup differences did not reach statistical significance, it was found that GNC participants reported the highest prevalence at 39.7% followed by TM at 34.8% and TW at 30.1% [61].

We located a limited number of studies investigating internet addiction among TGD communities, none of which indicated that transgender populations are at an elevated risk compared to their cisgender counterparts. For instance, a nationwide sample of U.S. college students determined that 14.29% of GMs reported past year problematic internet use compared to 14.05% of cisgender men and 9.15% of cisgender women [62]. Another study measured the average PIU (problematic internet use) score among transgender and cisgender subgroups [63]. While this study did not report prevalence statistics, it ultimately concluded that average PIU scores did not differ significantly as a function of gender identity.

The prevalence of sex addiction among TGD communities also has been investigated. One study concluded that 15.1% of transgender adults had experienced a lifetime sex addiction [55]. Although no large-scale epidemiologic studies have been performed, older studies indicate that the prevalence of sex addiction among the national population ranges from 3–6% [64,65]. We located one study that examined compulsive sexual behavior among 1401 adults stratified into two cisgender and two transgender subgroups [66]. This particular study, which assessed participants using the Hypersexual Behavior Inventory (a questionnaire with strong concurrent validity with other measures of sex addiction), determined that 19.2% of individuals met the criteria for compulsive sexual behavior. After stratifying participants into their gender subgroups, it was concluded that cisgender males reported the highest rates of hypersexuality at 29.3% followed by TW (8.7%), cisgender women (8.2%), and TM (6.2%). However, it should be noted that this inventory has not been validated on the TGD population.

We also identified a limited number of studies that examined the risk of exercise addiction among TGD individuals. A study of 484 transgender adults concluded that rates of past month excessive exercise (defined as exercising in a driven or compulsive way as a means of controlling weight, shape or amount of fat, or burning off calories) were virtually the same between TM and TW (8.0% vs. 8.1%) [60]. We also located a number of recent studies that investigated the susceptibility of TM to exercise addiction [67,68]. More specifically, these studies pointed out that TM suffer from high rates of muscle dysmorphia, which is highly associated with compulsive exercise. While we did not locate any studies that demonstrated higher rates of exercise addiction among TM relative to TW, future research should focus on the potential vulnerability of this GM subgroup who might use exercise addiction to combat feelings of gender dysphoria around soft and/or curvy parts of their bodies.

Of the remaining behavioral addictions (work, shopping, and love), the current literature was extremely sparse. We identified one study that determined that 17.8% of transgender participants met criteria for a lifetime work addiction while 37.0% had experienced a lifetime shopping addiction [55]. It should be noted that this study only examined 73 transgender adults and did not include a cisgender reference group for additional context. We did not locate a single study that examined the prevalence of love addiction as a function of gender identity, though relatively few studies of love addiction exist on any population.

## 4. Discussion

Through our review of the literature, we identified a comprehensive body of research that analyzed addiction rates among TGD individuals as a single, non-stratified group. The vast majority of these studies reached a similar conclusion: the prevalence of addiction is significantly higher in transgender populations compared to cisgender populations. While we did locate a few studies where prevalence rates were not statistically different between the two populations, and some studies with contradictory results, these other studies were perhaps even more illuminating because they highlighted alternative disparities that are often overlooked in traditional prevalence studies. For instance, a study measuring the prevalence of heavy episodic drinking concluded that transgender participants actually reported lower rates than both cisgender men and cisgender women [20]. However, when measuring the prevalence of alcohol-related problems (i.e., suicidality, sexual assault, etc.) among the same sample, transgender participants reported higher rates than either of the cisgender reference groups. Another study reached the conclusion that alcohol consumption did not vary significantly between transgender and cisgender subgroups; however, when researchers measured an individual’s motivation for drinking, it was found that transgender participants were significantly more likely to identify negative reasons (i.e., stress reduction, social anxiety, and self-esteem issues), while cisgender participants were much more likely to drink for positive social reasons (i.e., to have a good time with your friends and to celebrate) [30]. Although these types of studies were sparse, they provide additional context for the susceptibility of transgender populations to the downstream effects of addiction and highlight additional disparities that extend beyond traditional prevalence statistics.

While there is a strong body of research comparing addiction rates between transgender and cisgender populations, the core purpose of our paper was to expand upon the empirically accepted conclusion that transgender individuals are, as a whole, at increased risk of addiction. Therefore, we focused on establishing the addiction discrepancies that exist between individual TGD subgroups. We identified a growing number of studies that compared addiction rates between TW and TM. Through our analysis of these studies, several differences in prevalence rates became apparent. Most notably, TM appeared to be at a higher risk than TW for most SUD categories (i.e., AUDs, NUDs, and CUDs). However, in our review of illicit drug use, it became evident that TW may be uniquely susceptible to methamphetamine use as well as injection drug use; the latter being particularly concerning given its association with parenterally transmitted infections such as HIV and Hepatitis C.

Through our literature review, we identified a number of high-powered studies that analyzed a single TGD subgroup without the inclusion of a reference group. Among these single subgroup studies, the majority focused specifically on the TW community, with only a few studies that were dedicated exclusively to TM populations. Interestingly, in the studies that did compare addiction prevalence rates between TGD subgroups, there was more evidence to suggest that TM were at a higher risk for most forms of addiction (i.e., alcohol, nicotine, and cannabis) than TWs. While it is undeniable that TW face unique obstacles that intersect with their substance use (i.e., risky sexual behavior, suicidality, and sexual assault), future research should also investigate the distinctive reasons that underlie substance abuse among TM. After all, establishing a deeper understanding of a subgroup’s unique predispositions has clinically important implications for screening at-risk patients and tailoring treatments.

Our review also included numerous studies that examined addiction rates among nonbinary individuals, a large portion of the TGD population that remains largely understudied. While we identified studies suggesting that nonbinary and GNC individuals may be at a higher risk for certain addictions relative to their transgender counterparts (i.e., AUDs), the majority of current research suggests that their addiction rates do not significantly differ from TW and TM. In fact, some studies even indicated that nonbinary and GNC individuals may be at a lower risk for certain addictions compared to the other TGD subgroups (i.e., NUDs). However, it should be noted that the number of studies that stratified participants into separate nonbinary or GNC subgroups was sparse, and as a result, it is difficult to draw any overarching conclusions.

As in our previous review of SM subgroups, there was substantially less research that investigated behavioral addictions as a function of gender identity [1]. However, despite a limited number of published studies, there is compelling evidence to suggest that prevalence rates may vary significantly between individual GM subgroups. For instance, we found one study indicating that a random sample of TM reported levels of body dysmorphia comparable to those with a diagnosed eating disorder [68]. A similar study concluded that TM, as a result of their elevated rates of muscle dysmorphia, were more likely to engage in compulsive exercise [69]. We also located a study that indicated that nonbinary and GNC individuals had higher rates of binge eating, and therefore, were at a higher risk of food addiction relative to both TM and TF. While there is a clear need for more studies that characterize behavioral addictions among the TGD community, there are also a number of unique concerns that make this area of research particularly challenging. After all, PG is currently the only behavioral addiction with diagnostic criteria included in the DSM-V. Of the remaining behavioral addictions, researchers utilized differing sets of diagnostic criteria that may not measure the same addiction outcome. As a result, this absence of a single, uniform set of diagnostic criteria makes it particularly challenging to compare results from one study to another. It has also been purported that stigmatization of transgender populations may result in provider bias that ultimately impacts prevalence estimates. In fact, one study suggested that transgender individuals, when compared to cisgender populations, are less likely to receive a diagnosis of sex addiction as a result of the general perception of sex positivity among the LGBTQ community [70]. While future studies are likely to provide further insight into the susceptibility of TGD subgroups to behavioral addictions, it is also important that future researchers acknowledge and account for the inherent challenges associated with this area of study.

Through our analysis of TGD addiction rates, we also identified several important yet overlooked areas of research that deserve additional investigation. For instance, while we only identified a limited number of studies that examined co-occurring addictions as a function of gender identity, it became apparent that substantial differences existed between various addiction subtypes. For example, two studies, which examined the co-occurrence of non-medical prescription opioid (NMPO) use with other substances, concluded that TGD individuals with a history of NMPO use were over twice as likely to report both cannabis and nicotine use; however, rates of alcohol use remained unchanged as a function of NMPO use [28,41]. Several studies also investigated whether HIV status was significantly associated with a transgender individual’s risk of abusing a particular substance. As demonstrated in the co-occurring addictions example, associations differed markedly based on which addiction subtype was being analyzed. For example, TW who were HIV+ were more likely to use methamphetamine, while those who were HIV- were more likely to use alcohol; additionally, cannabis use remained unchanged regardless of one’s HIV status. Through understanding the degree to which certain addictions co-occur with other addictions or disease states, clinicians may be more effective in their efforts to screen at-risk GM patients and provide appropriate interventions.

In addition to investigating co-occurring addictions, future studies should also consider additional methods of stratification when analyzing TGD populations. We located one study that stratified TGD participants into two groups based on their sexual identity (i.e., heterosexual vs. LGBQ) [62]. This particular study, which measured the prevalence of illicit drug use, concluded that heterosexual transgender individuals had a lower risk of use relative to their TGD SM counterparts. While heterosexuality has been widely accepted as a protective factor for cisgender populations, there is effectively no research examining the impact of sexual identity within the TGD community. Additionally, visual conformity (i.e., the degree to which transgender individuals are outwardly perceived as their affirmed gender) represents a TGD-specific correlate that likely influences the TGD community’s predisposition to various forms of addiction. In fact, one study in particular determined that low levels of visual conformity increased one’s odds of nicotine use [37]. The same study also determined that participants who had disclosed their trans identity were more likely to report nicotine use. Expanding the stratification of TGD to include dimensions other than gender identity could pinpoint a new group of highly at-risk TGD individuals that have been overlooked by traditional SM categorizations.

It is important to note that one of the predominant reasons that TGD individuals experience higher rates of mental health challenges, addictions, and other presumptively negative outcomes is that of minority stress and stigmatization [71]. Living in a world where one experiences discrimination, harassment, and violence at interpersonal, institutional, and even ideological levels can result in TGD individuals seeking out additional coping mechanisms as compared to their cisgender counterparts [72]. Simply being TGD by itself likely does not indicate an increase predisposition to addictive behavior; however, living in a hurtful and traumatizing world can, indeed, result in such behaviors. Understanding how to support individuals within this framework is key for clinicians in engaging their TGD clients and patients around addiction [73].

It should also be noted that the vast majority of scales and instruments that are used to assess for addiction-related behaviors have not been validated on TGD participants, and rather, are often validated solely on cisgender individuals. In order to ensure these scales accurately measure what they purport to measure, future research must be conducted to norm these existing instruments on this population and/or create new instruments that can accurately measure addictive behaviors among this population. Similarly, the criteria for viewing some addictive behaviors can be viewed as transphobic, such as gendered rates for what is considered binge drinking, when there is no number of drinks noted for nonbinary individuals, and transgender women might be significantly taller and/or weigh more than cisgender women (and vice versa for transgender men and cisgender men). Both research and practice must take these challenges into account when engaging TGD individuals.

## 5. Conclusions

It is widely established in the literature that TGD individuals are at an elevated risk for developing various addictions. Given that the majority of published studies analyze TGD individuals as a single homogenous group rather than by within-group genders, there is less understanding of the differences that exist between individual TGD subgroups. Our review is unique in that it investigates how differences in gender identity may ultimately translate into significantly different addictive behaviors. By understanding the addiction predispositions of individualized TGD subgroups and causal etiologies of these subgroups (e.g., stigma-related trauma and lack of coping outlets), clinicians may be better equipped to provide earlier interventions and more tailored treatments, ultimately leading to reduced healthcare expenditures. Our review also adds to the literature by assessing both SUDs and behavioral addictions as functions of gender identity.

## Figures and Tables

**Figure 1 ijerph-18-08843-f001:**
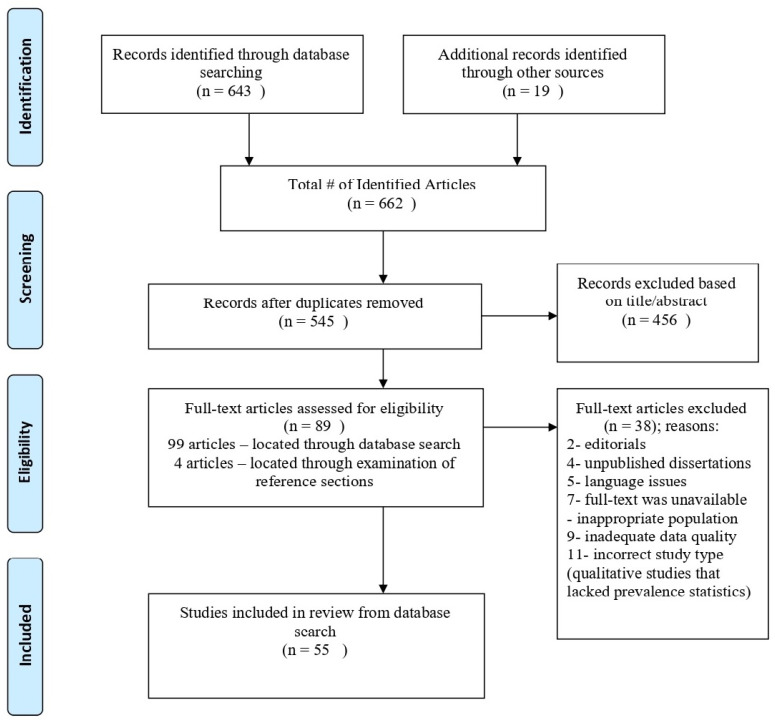
Diagram of study identification and selection process.

## Data Availability

Not applicable.

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
