# Peer review of "Review: Prevalence of Addictions among Transgender and Gender Diverse Subgroups"

_ijerph, 2021, doi:10.3390/ijerph18168843_

Round 1
Reviewer 1 Report
The present review identified interesting results regarding the higher risk of additions in TGD individuals, and the lack of research in substance and behavioral additions that incorporated subgroups of TGD individuals.
This review is organised in a nice, easy-to-read manner. However, it would be better if Tables/Figures could be provided to compare the key findings for each addiction between transgender and cisgender groups and between subgroups of GM and SM groups, with citations.
Systematic? Did this systematic review follow Preferred Reporting Items for Systematic Reviews and Meta-Analyses (PRISMA)? If not, why?
Is this review registered at PROSPERO? If not, why?
Did studies have to be written in English to be included? If not, who translated them for your research? And information about sample size in literature should be reported, is there a threshold of sample size as an inclusion criterion?
How many studies relating to pathological gambling have been identified and included? Page 10: findings about gambling only have one citation. Also, "problem gambling" should have been included in search terms.
Author Response
The present review identified interesting results regarding the higher risk of additions in TGD individuals, and the lack of research in substance and behavioral additions that incorporated subgroups of TGD individuals.
This review is organised in a nice, easy-to-read manner. However, it would be better if Tables/Figures could be provided to compare the key findings for each addiction between transgender and cisgender groups and between subgroups of GM and SM groups, with citations.
I would be glad to generate a table that summarizes each study and allows for comparison, but would need more than a 5 day turnaround period to complete it (as I’m currently working full time on a medical service and have limited free time).
Systematic? Did this systematic review follow Preferred Reporting Items for Systematic Reviews and Meta-Analyses (PRISMA)? If not, why?
Thank you for pointing this out. Our paper is a scoping review so Preferred Reporting Items for Systematic Reviews was not followed. We utilized a scoping review model instead. I recognize that I used the phrase “systematically” in the abstract which I have since removed. I have also more clearly defined throughout the manuscript that our paper is a scoping review.
Is this review registered at PROSPERO? If not, why?
We support future work doing a more in-depth look at this comparison, but that it is outside the scope of this current project.
Did studies have to be written in English to be included? If not, who translated them for your research? And information about sample size in literature should be reported, is there a threshold of sample size as an inclusion criterion?
Thank you for pointing this out. I’ve updated my inclusion and exclusion criteria to reflect these points.
How many studies relating to pathological gambling have been identified and included? Page 10: findings about gambling only have one citation. Also, "problem gambling" should have been included in search terms.
Yes, we were only able to locate one relevant study that analyzed pathological gambling as a function of gender identity. I changed the wording of the intro sentence to acknowledge that the research is “limited” as opposed to “growing.”
Reviewer 2 Report
This very interesting study, reviews LGBTQ subsets vulnerability to substance abuse. The Introduction reports on the various statistics regarding suicidality, discrimination and poorer health outcomes for these groups and finally focuses on the prevalence and substance abuse and addiction occurring in transgender males and females. The article is of high standard and very well written.
- Originality/Novelty: Although the topic is not original or novel, it is a “review”, I found the study very interesting and unique from the perspective of the focus of the study group, ie., transgender males and females.
- Significance: The significance of the study is timeous and the conclusions are justified and supported by the results.
- Quality of Presentation: The article is written in academic language and includes a figure that enhance the presentation of the study. The Abstract contains abbreviations that are well clarified early on that makes reading the rest of the article easier.
- Scientific Soundness: The analysis is clearly defined and well set out and the study appears to be scientifically sound.
- Interest to the Readers: The article would interest the readers of IJERPH as the empirical and theoretical contribution of the content is aligned with the aims and scope of the journal. I found the article of prime interest, especially during the current times we live in.
- Overall Merit: The study can be published
- English Level: Good
References
- In-text references mainly comply with journal requirements.
- Sufficient current studies and citation were used.
- The Reference list do not seem to conform to journal requirements and appears to be in APA style.
Strengths of the paper
- Methodology and selection of participants are swell explained and sample size of articles reviewed is sufficient.
- Highlights a vulnerable group within the LGBTQ community.
- Highlights areas for further research.
- The main strength of this paper lies in the “Discussion” section which is comprehensive and thoroughly addressed.
Weakness of the paper
- None obvious weaknesses.
Major Points
- Please check Reference List for style
Minor Points
- Points 3 – 7 should be sub-headings and perhaps include a main heading, ie., “Results”. Initially I suspected that it belonged under “Introduction”, but later realised it refers to “Results”. It is not a critical issue, merely an observation.
Author Response
Major Points
- Please check Reference List for style
I completed my references based on the IJERPH website which stated “our references may be in any style, provided that you use the consistent formatting throughout.” I’m happy to redo my references if this does not apply.
Minor Points
- Points 3 – 7 should be sub-headings and perhaps include a main heading, ie., “Results”. Initially I suspected that it belonged under “Introduction”, but later realised it refers to “Results”. It is not a critical issue, merely an observation.
Thank you for pointing this out. I agree. Result header was added.
Reviewer 3 Report
- At line 526 there is a blank that probably means that something is missing that should be there.
- On page 13, the discussion of sexual minority theory is rather limited; Bailey et al. (2016) have criticized it for such reasons. For example, how do we know if perhaps TGD persons are rejected because of higher rates of dysfunctional behaviors or that perhaps there is an iterative pattern here (reciprocal influences? Do cisgender persons who use illegal drugs, for example, experience rejecting microaggressions from others in similar ways as TGD persons? Maybe a study has compared using/non-using cisgender and TGD persons to try to tease out such issues.
- Overall, a very well done paper.
Author Response
At line 526 there is a blank that probably means that something is missing that should be there.
Thank you for pointing out this type. Change made.
On page 13, the discussion of sexual minority theory is rather limited; Bailey et al. (2016) have criticized it for such reasons. For example, how do we know if perhaps TGD persons are rejected because of higher rates of dysfunctional behaviors or that perhaps there is an iterative pattern here (reciprocal influences? Do cisgender persons who use illegal drugs, for example, experience rejecting microaggressions from others in similar ways as TGD persons? Maybe a study has compared using/non-using cisgender and TGD persons to try to tease out such issues.
This is a very interesting point. And I agree that minority stress theory is not all encompassing in any way (especially as it relates to TGD populations). Unfortunately, we were unable to find any studies that directly investigated differences in use patterns among cisgender and TGD individuals. But I agree with your critique and think this topic is an important area for future research.
Overall, a very well done paper.